# Whole Cow’s Milk but Not Lactose Can Induce Symptoms in Patients with Self-Reported Milk Intolerance: Evidence of Cow’s Milk Sensitivity in Adults

**DOI:** 10.3390/nu13113833

**Published:** 2021-10-27

**Authors:** Antonio Carroccio, Maurizio Soresi, Beatrice Mantia, Francesca Fayer, Francesco La Blasca, Aurelio Seidita, Alberto D’Alcamo, Ada Maria Florena, Chiara Tinè, Chiara Garlisi, Pasquale Mansueto

**Affiliations:** 1Unit of Internal Medicine, “V. Cervello” Hospital, Ospedali Riuniti “Villa Sofia-Cervello”, 90146 Palermo, Italy; 2Department of Health Promotion Sciences, Maternal and Infant Care, Internal Medicine and Medical Specialties (PROMISE), University of Palermo, 90127 Palermo, Italy; 3Unit of Internal Medicine, Department of Health Promotion Sciences, Maternal and Infant Care, Internal Medicine and Medical Specialties (PROMISE), University of Palermo, 90127 Palermo, Italy; maurizio.soresi@unipa.it (M.S.); beatrice.mantia@alice.it (B.M.); francesca.fayer@gmail.com (F.F.); francescolablasca@gmail.com (F.L.B.); xenis86@gmail.com (A.S.); adalcamo@hotmail.it (A.D.); chiara.tine@community.unipa.it (C.T.); mchiaragarlisi@gmail.com (C.G.); pasquale.mansueto@unipa.it (P.M.); 4Unit of Pathology, Department of Health Promotion Sciences, Maternal and Infant Care, Internal Medicine and Medical Specialties (PROMISE), University of Palermo, 90127 Palermo, Italy; adamaria.florena@unipa.it

**Keywords:** self-reported milk intolerance, lactose intolerance, lactose hydrogen breath test, cow’s milk protein allergy, IBS, HLA, duodenal histology

## Abstract

Background: Lactose intolerance is the most frequent food intolerance, but many subjects with self-reported milk intolerance (SRMI) are asymptomatic at lactose hydrogen breath test (LHBT). The aim of this study was to evaluate the frequency of lactose intolerance in SRMI patients and their clinical characteristics. Methods: In a retrospective study, the clinical records of 314 SRMI patients (259 females, mean age: 39.1 ± 13.5 years) were reviewed; 102 patients with irritable bowel syndrome (IBS) served as controls. In a prospective study, 42 SRMI patients, negatives at the LHBT, underwent a double-blind, placebo-controlled (DBPC) whole cow’s milk challenge. Results: In the retrospective study, only 178 patients (56%) were lactose maldigesters and intolerant at LHBT; 68% of the subjects with SRMI were suffering from IBS; 74% reported dyspepsia (*p* = 0.0001 vs. IBS controls); and weight loss was recorded in 62 SRMI patients (20%) (*p* = 0.01 vs. IBS controls). Duodenal histology showed intra-epithelial lymphocytosis in about 60% of cases. In the prospective study, 36 patients (86%) experienced symptoms during the DBPC cow’s milk challenge, and only 4 patients (9%) reacted to placebo (*p* = 0.0001). Conclusions: A percentage of SRMI patients were not suffering from lactose intolerance. DBPC revealed that SRMI patients had clinical reactions when exposed to whole cow’s milk.

## 1. Introduction

Cow’s milk and dairy products are among the foods most frequently avoided in the general population, as a great percentage of people consider that ingesting these kinds of foods causes gastrointestinal and extra-intestinal symptoms, especially those affected by irritable bowel syndrome (IBS) [1].

Although milk and its derivatives are FODMAP-rich foods, they could have a different role in determining gastrointestinal symptoms. In fact, lactose malabsorption is caused by low lactase levels; this “hypolactasia” is genetically determined in adults and is related to ethnicity, the rates of lactose malabsorption being highest in African Americans (60–100%) and progressively decreasing in the Mediterranean area (40%) and Northern Europe (10–20%) [2]. Due to this very high frequency of lactase deficiency, the presence of abdominal pain, bloating, flatulence, and diarrhea after milk ingestion is almost automatically considered to indicate a diagnosis of “lactose intolerance”. However, it should be underlined that many patients who self-report symptoms after milk ingestion and consequently avoid milk consumption are found to be asymptomatic at the hydrogen breath test for lactose (LHBT), which is currently considered the “gold standard” to diagnose lactose “maldigestion” and lactose “intolerance” [3,4]. Consequently, in these subjects, many other differential diagnoses should be considered [3,5]. Furthermore, no data are available to ascertain whether whole milk consumption can determine symptoms in SRMI patients who are asymptomatic during LHBT.

The aims of the present study were to evaluate: (a) the frequency of lactose intolerance at LHBT in SRMI patients; (b) the clinical characteristics of patients with SRMI, compared with patients suffering from IBS unrelated to food intolerance; and (c) the clinical reaction to a double-blind, placebo-controlled (DBPC) challenge with whole cow’s milk or placebo in patients with SRMI and negative LHBT.

## 2. Materials and Methods

This study included two different parts, a retrospective and a prospective one.

(A)Retrospective Study

In the retrospective part of the study, we reviewed the clinical charts of all of the patients with self-reported lactose or milk intolerance who attended the outpatient centers of the Department of Internal Medicine at the University Hospital of Palermo (Italy) or of the Department of Internal Medicine at the Hospital of Sciacca (Agrigento, Italy) between January 2011 and June 2019. The clinical charts included specific sections for clinical presentation, results (hydrogen excretion) and clinical symptoms on LHBT, associated functional gastrointestinal diseases (dyspepsia and/or irritable bowel syndrome), associated autoimmune diseases, associated self-reported, non-celiac wheat sensitivity (NCWS), and associated multiple food sensitivities. In a percentage of the patients, we also recorded and analyzed the presence of the HLA DQ2 and DQ8 haplotypes and the results of a duodenal histology study. The clinical charts used in the present study had been previously validated in retrospective studies on NCWS by our group [6,7]. Incomplete clinical charts were excluded. In addition, we selected a control group of 102 patients with IBS, 90 of them with IBS-diarrhea and 12 with alternate bowel movements; they were randomly chosen by a computer-generated method from subjects diagnosed during the same period (2011–2019) and were age- (±2 years) and sex-matched (±5%) with the patients with SRMI. The IBS controls had been receiving an elimination diet excluding cow’s milk and wheat during their diagnostic work-up in our center and had not shown any clinical improvement. Furthermore, they underwent the identical diagnostic work-up to the patients with SRMI in order to exclude “organic diseases” (see Appendix A, below “Other investigations”).

The inclusion criteria were: (a) SRMI and the consequent exclusion of milk from the diet for one year or more; (b) reporting of a regular recurrence of symptoms after involuntary milk consumption; (c) LHBT performed in our centers; (d) a one-year follow-up, including at least two visits to our centers after LHBT; and (e) age between 18 and 65 years.

The exclusion criteria were: (a) concomitant diagnoses of “organic” gastrointestinal diseases; and (b) current gastrointestinal infections and/or current use of antibiotics, probiotics, steroids, and immunosuppressant treatment.

For details on hydrogen breath test for lactose, other performed assays, and definitions, see Appendix A) “Prospective Study. DBPC challenge with whole cow’s milk.”

(B)Prospective Study. DBPC challenge with whole cow’s milk.

In the prospective part of the study, a DBPC whole cow’s milk challenge was proposed to the SRMI patients who had tested negative and were asymptomatic at the LHBT. These patients belonged to the same group included in the retrospective part of the study. The DBPC challenges were performed at the outpatient center of the Department of Internal Medicine at the University Hospital of Palermo (Italy) between July 2019 and January 2020.

Among those who gave their informed consent, 42 patients (32 F, mean age ± SD: 33.4 ± 11.7 years) were randomized to undergo the DBPC challenge. For DBPC whole cow’s milk challenge details, see Appendix A.

### Statistical Analysis

Data were expressed as mean ± SD when distribution was Gaussian, and in order to evaluate differences in mean values between groups, the Student’s t-test was used. Comparisons between more than 2 groups were performed by ANOVA, followed by post hoc analysis using Bonferroni tests. Otherwise, data were expressed as median and range and analyzed with the Kruskal–Wallis and the Mann–Whitney U tests. The chi-square test and the Fisher exact test were used to compare frequency values in the various study population groups.

All analyses were performed using the SPSS software package (released in 2007, version 16.0., Chicago, IL, USA).

The study protocol conformed to the ethical guidelines of the Declaration of Helsinki and was approved by the Human Research Committee of the University Hospital of Palermo (record number 10/2019) and registered at clinicaltrials.gov (registration number: NCT03008252). All subjects consented to participate in the study.

## 3. Results

(A)Retrospective Study

One thousand two hundred and twenty patients with SRMI underwent LHBT during the study periods. Of these, only 314 (259 females, mean age: 39.1 ± 13.5 years) entered the present study, as all of the others met the various exclusion criteria. Appendix A summarizes the flow chart of the study and the reasons for exclusion.

Figure 1 shows the classification of the patients according to the LHBT results. Only 178 patients (56%) were lactose maldigesters and intolerant (i.e., positive LHBT and symptomatic during or within 24 h after the test). One hundred patients (32%) did not show any symptoms during and after the LHBT and were considered “tolerant”. Among these, 82 subjects were negative (“digesters”), and 18 were positive (“maldigesters”); they were grouped together as “lactose tolerant” in the subsequent tables and analysis. Thirty-six patients (12%), despite being LHBT negative (“digesters”), reported symptoms during or after the assay and were grouped separately as “digesters intolerant”.

Table 1 shows the demographic and clinical characteristics of the study patients. Patients of all of the three SRMI groups reported diarrhea and abdominal pain as their most common gastrointestinal symptoms due to milk ingestion. However, 15–19% of the patients reported constipation. A certain percentage of patients also self-reported extra-intestinal symptoms following milk ingestion: fatigue (5% of cases) and sweating (4%) were the most frequent among the “lactose tolerant”.

Table 2 shows the frequency of IBS, dyspepsia, anemia, weight loss, and autoimmune diseases in the three SRMI groups and in the IBS controls. Two hundred and fourteen (68%) of the SRMI subjects were found to have IBS, with a similar frequency in the tolerant and intolerant groupings at the LHBT. Two hundred and thirty-two SRMI patients (74%) reported dyspepsia, and all three SRMI groups showed a higher frequency of dyspepsia than the IBS controls (*p* = 0.0001, 0.0001, and 0.005, respectively). Weight loss was recorded in 62 patients with SRMI (20%), with a frequency significantly higher than in the IBS controls (*p* = 0.01). Both the lactose tolerant and lactose intolerant subjects at the LHBT had a higher frequency of weight loss than the IBS controls. Anemia was observed in 45 SRMI patients (14%); its frequency was significantly higher in the group of subjects with SRMI who were asymptomatic at the LHBT than in the other groups of SRMI patients and the IBS controls.

Considering the whole group of SRMI patients, the clinical charts included the results of the HLA DQ2 and DQ8 haplotypes in 176 cases and of the duodenal histology study in 98 cases (see Table 3). Ninety-six out of 176 (54%) subjects with SRMI carried the HLA DQ2 or DQ8 allele, with a frequency significantly higher than in IBS controls (*p* = 0.01). In general, all the subgroups of SRMI, whether lactose tolerant or intolerant at the LHBT, tended to show a higher frequency of haplotypes DQ2 and DQ8 than IBS controls, but the differences were not significant (for “lactose tolerant” vs. IBS, *p* = 0.07). Duodenal histology showed a normal villus/crypt ratio (>3) in all of the SRMI patients. However, the frequency of intra-epithelial lymphocytosis was high in all SRMI study groups, with values between 57% and 70% of cases (60% in the whole group). Fifteen of the 314 (4.7%) SRMI patients tested positive for serum specific IgE and/or skin prick test for cow’s milk proteins. Analysis of the subgroups showed the highest frequency of positive IgE-mediated tests in the subjects who were “tolerant” at the LHBT, and this was significantly higher than in the subgroup of “intolerant” patients (*p* < 0.05). Finally, a concomitant self-reported NCWS was recorded in 101 of the 314 (32%) SRMI patients, with 24 (8%) reporting multiple food intolerance.

(B)Prospective Study. DBPC challenge with whole cow’s milk.

Forty-two SRMI patients who tested negative and were asymptomatic at the LHBT were randomized to the DBPC whole cow’s milk re-challenge study. Twenty-two patients received cow’s milk and 20 rice milk at the first challenge. They were all asymptomatic before undergoing the study. The second challenge with rice or cow’s milk, respectively, was performed one week after the first one and in all cases when the patients were asymptomatic and on a cow’s milk-free diet. Thirty-six (86%) experienced symptoms again (increase in VAS score > 30 of at least one of the symptoms) during the DBPC cow’s milk challenge. The clinical reactions to cow’s milk ingestion occurred after a median time of 2 h (range: 30 min to 6 h). Only four patients (9%) reacted to the rice milk challenge (placebo) (*p* = 0.0001 (Fisher test) vs. cow’s milk challenge). Excluding the four patients who also reacted to placebo, 32 of the 42 (76%) patients were confirmed to be suffering from cow’s milk intolerance but not lactose intolerance, as evaluated by the DBPC milk challenge.

Table 4 shows the number of bowel movements and the severity of the symptoms during the 24 h after the challenges. Both the number of the bowel movements (*p* = 0.002) and the severity of the intestinal symptoms, such as the overall symptoms (*p* = 0.0001), were significantly higher after the cow’s milk than after the rice milk challenge.

## 4. Discussion

Lactose intolerance is the most frequent food intolerance worldwide. In fact, its prevalence was estimated at 68% in the general world population [8]. However, this very high percentage does not correspond to an identical frequency of lactose intolerance, as several factors are known to contribute to the appearance of symptoms after lactose ingestion in subjects with lactose maldigestion. Among these factors is a concomitant condition of IBS, probably secondary to the visceral hypersensitivity which characterizes the disease. Consequently, SRMI is very frequent in IBS patients, and lactose intolerance is often diagnosed independently of the results of a LHBT.

In the retrospective part of the present study, we included a large group of subjects with SRMI. It must be underlined that a high percentage of the study patients were suffering from functional gastrointestinal complaints. IBS, in fact, was found in about 70% and functional dyspepsia in 74% of our patients, the latter being more frequent in the SRMI than in the IBS controls. A very high frequency of association between functional gastrointestinal complaints and lactose intolerance has been previously reported in the literature [4,8].

Of the whole group of 314 SRMI patients, only 178 (56%) showed symptoms during or within 24 h after a positive LHBT (i.e., lactose maldigesters and intolerant). The lack of correspondence that we found between the presence of self-reported symptoms on milk ingestion and the results after lactose load at the LHBT in a variable percentage of cases has been reported in previous studies [9,10,11,12,13,14,15], with a wide percentage range of positive LHBT in SRMI of between 4% and 86% [16]. This lack of correspondence is highest in patients with functional gastrointestinal problems, i.e., IBS and functional dyspepsia [15], as demonstrated by the very high frequency of SRMI in European patients with IBS and the low rate of genetic/genetically determined lactase non-persistence [1].

Several explanations could be proposed for this lack of correlation. Among these, there may be a simple nocebo effect when patients eat dairy products or upon the concomitant consumption of other FODMAP-rich foods, which can contribute to determining symptoms. However, these plausible hypotheses do not seem to consider that milk is not “only lactose” and that other pathogenetic mechanisms should be considered. In this respect, our study highlighted some interesting clinical and laboratory characteristics. We included SRMI patients (a high percentage of whom were suffering from functional gastrointestinal diseases) who had regularly reported symptoms on involuntary milk ingestion and, consequently, had avoided dairy products for one year or more. Our strict inclusion criteria, therefore, probably selected subjects with “real milk-related” symptoms.

A first observation from our cohort data is that about 20% of the SRMI patients (15% of the “lactose intolerant” and 18% of the “lactose tolerant”) reported constipation following milk ingestion. Constipation cannot be a consequence of lactose maldigestion, although it should be remembered that it can be included among the symptoms caused by cow’s milk protein allergy in pediatric patients [17,18,19,20] and has also been reported in adults [21,22]. Furthermore, the extra-intestinal symptoms, frequently reported by SRMI patients after milk ingestion, remain unexplained [4]. Among these, we recorded fatigue and headache, which have also been reported by patients with self-reported NCWS, another condition with a still undefined pathogenesis [23] that overlaps with IBS [24].

In addition, we observed systemic symptoms in our cohort. Weight loss was recorded in 20% of the SRMI patients, with a frequency significantly higher than in the IBS controls (*p* = 0.01), and this occurred both in the lactose tolerant (24%) and the lactose intolerant (18%) subjects at the LHBT. Furthermore, 14% of the SRMI patients had anemia, which was more frequent (25%) in the subgroups of the lactose tolerant at the LHBT and showed significantly higher frequency than in the IBS controls. Both these symptoms could be related to dietary self-restriction, to which patients very often resort in the attempt to limit their problems. An elimination diet can obviously determine malnutrition. However, the hypothesis of concomitant, subtle, intestinal malabsorption cannot be excluded.

In this respect, another interesting result came from the duodenal histology data. In fact, 60% of the patients had intra-epithelial lymphocytosis (IEL count > 25/100 EC). Very interestingly, recent data obtained with the confocal endomicroscopy method demonstrated that a percentage of IBS patients, often with self-reported food intolerances, reacted to the instillation of milk into the duodenum with the appearance of intercellular lesions and tight junction breaking and with an immediate increase in intra-epithelial lymphocyte infiltration into the mucosa [25,26]. Our data on the inflammatory infiltration of the duodenal mucosa could thus suggest that a percentage of patients with SRMI could be suffering from “cow’s milk protein hypersensitivity”, with a pathogenetic model similar to that proposed for patients suffering from NCWS [23,26]. In this respect, the concomitant condition of self-reported NCWS observed in one-third of the study patients is in keeping both with our experience of a frequent association between NCWS and cow’s milk protein hypersensitivity [6,23,27] and with the results of endomicroscopy studies in IBS patients, which showed a multiple food “reaction” in almost all of the subjects studied [25,26].

A possible role for the alternative “immunologic-inflammatory” hypothesis in a percentage of our patients with SRMI could also be suggested by the high frequency of the DQ2/DQ8 haplotypes (54%) which they carried. It is interesting to note that in the SRMI patients neither the frequency of increased duodenal IELs (Marsh 1 lesion) nor of DQ2/DQ8 HLA haplotypes was related to a concomitant condition of self-reported NCWS. In fact, similar frequencies of these two parameters were observed in patients with associated self-reported NCWS and in those who did not report this association (data not shown).

The alternative hypothesis—that other milk components may cause symptoms in SRMI patients—was confirmed by the prospective part of our study. In fact, 42 SRMI patients who had tested negative and were asymptomatic at the LHBT underwent a DBPC challenge with whole cow’s milk or rice milk (placebo). The challenge showed that 32 of the 42 (76%) patients suffered from cow’s milk intolerance but not lactose intolerance. Furthermore, the number of bowel movements and the severity of the intestinal and of the overall symptoms (*p* = 0.0001) were significantly higher after the cow’s milk than after the rice milk challenge.

Despite the intriguing and innovative data obtained from the study of our cohort, the limitations of the research must be underlined. First, a large part of our data came from a retrospective study in which we included only one-quarter of over 1200 patients who underwent LHBT during the study period. Although a percentage of the cases were excluded for a “correct reason” (i.e., non-hydrogen producing patients), most had incomplete records, and we cannot exclude the possibility that the most accurate records and the most regular follow-ups were linked to the most complicated cases. This could have led us to overestimate the “inflammatory aspects” of the SRMI group. Second, our study was performed in two tertiary centers with experience in food-related diseases; consequently, it is possible that we observed a pre-selected population composed of patients who did not show a “simple” lactose intolerance as can be observed in the general population. Third, we did not perform a genetic assessment of lactase persistence; such data would have improved the study quality, although the lack of correlation between genotype and phenotype in the lactase intolerance setting is well known. Fourth, the patients who underwent HLA assay and duodenal biopsy could have been the most severe cases or those with a suspected concomitant celiac disease; thus, the high frequency of DQ2/DQ8 haplotypes and duodenal lymphocytosis which we reported could be due to the selection of cases, which did not reflect the total clinical records. Fifth, most of the patients included were females, and, therefore, these results should be considered limited to the female gender. However, we should recall that the female sex is also largely prevalent in patients suffering from NCWS, a condition which we consider very similar to that which we described in this study for a percentage of the SRMI patients included.

The strengths of the study, however, were the high number of patients included, the standardized method of performing the LHBT in only two centers, and the strict inclusion criteria. In fact, we studied SRMI patients who reported the regular recurrence of symptoms following involuntary milk consumption and had excluded milk and derivatives from their diet for one year or more; furthermore, a one-year follow-up, including at least two visits, after LHBT ensured the exclusion of diagnoses of other diseases. Moreover, we performed a prospective study on a percentage of the SRMI patients included in the retrospective phase of the study, which provided data of strong clinical reactions to whole cow’s milk during a rigorous DBPC study.

## 5. Conclusions

In conclusion, this study confirmed that a percentage of patients with SRMI were not suffering from lactose intolerance and showed that these subjects had clinical characteristics which suggested subtle malabsorption and immunologic activity in the intestinal mucosa. Furthermore, for the first time, we demonstrated that SRMI patients who tested negative at the HLBT showed a severe clinical reaction when exposed to whole cow’s milk during a DBPC challenge. This raises the hypothesis that non-IgE-mediated cow’s milk protein allergy, and not lactose intolerance, could cause symptoms in a still undefined percentage of SRMI patients.

## Figures and Tables

**Figure 1 nutrients-13-03833-f001:**
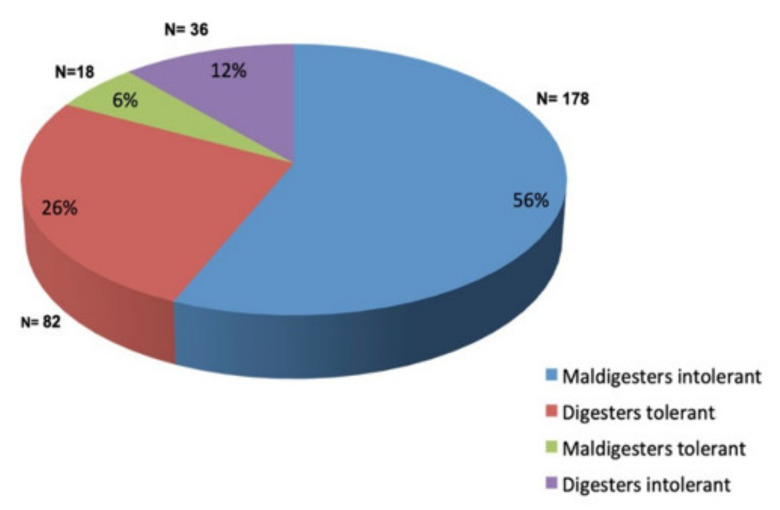
Classification of the patients according to the results of the lactose hydrogen breath test.

**Table 1 nutrients-13-03833-t001:** Demographic characteristics and self-reported gastrointestinal symptoms in the 314 study subjects, grouped according to the LHBT results as “Lactose tolerant”, “Lactose intolerant”, and “Lactose digesters/intolerant”, and in the IBS controls who did not report milk intolerance.

	Lactose Tolerant (n = 100) A	Lactose Intolerant (*n* = 178) B	Lactose Digesters/Intolerant (*n* = 36) C	IBS Controls (*n* = 102) D	*p*
Age (years) (X ± SD)	38.8 ± 13.6	39.4 ± 12.4	34.1 ± 11.2	38.1 + 13.2	NS
Women (number and %)	75 (75%)	154 (86%)	30 (83%)	40 (80%)	A vs. B 0.03
Symptoms self-reported as “caused by milk ingestion”					
Diarrhea	65 (65%)	110 (67%)	26 (72%)	NONE	NS
Constipation	18 (18%)	27 (15%)	7 (19%)	NONE	NS
Abdominal pain	65 (65%)	121 (68%)	56 (72%)	NONE	NS
Extra-intestinal symptoms	29 (29%)	32 (18%)	6 (17%)	NONE	NS

LHBT = lactose hydrogen breath test; IBS = irritable bowel syndrome; SD = standard deviation; NS = not significant.

**Table 2 nutrients-13-03833-t002:** Frequency of IBS, dyspepsia, anemia, weight loss, and autoimmune diseases in the 314 study subjects, grouped according to the LHBT results as “Lactose tolerant”, “Lactose intolerant”, and “Lactose digesters/intolerant”, and in the IBS controls who did not report milk intolerance.

	Lactose Tolerant (*n* = 100) A	Lactose Intolerant (*n* = 178) B	Lactose Digesters/Intolerant (*n* = 36) C	IBS Controls (*n* = 102) D	*p*
IBS	71 (71%)	116 (65%)	27 (75%)	102 (100%)	A vs. B NS A vs. C NS A vs. D = 0.0001 B vs. C NS B vs. D = 0.0001 C vs. D = 0.001
Dyspepsia	70 (70%)	136 (76%)	26 (72%)	40 (40%)	A vs. B NS A vs. C NS A vs. D= 0.0001 B vs. C NS B vs. D= 0.0001 C vs. D =0.01
Weight Loss	24 (24%)	32 (18%)	6 (16%)	4 (4%)	A vs. B NS A vs. C NS A vs. D = 0.001 B vs. C NS B vs. D = 0.01 C vs. D = 0.05
Anemia	25 (25%)	17 (9%)	3 (7%)	8 (8%)	A vs. B = 0.001 A vs. C = 0.05 A vs. D = 0.001 B vs. C NS B vs. D NS C vs. D NS
Presence of concomitant autoimmune diseases	15 (15%)	27 (15.2%)	4 (11.2)	10 (10%)	All the comparisons were NS

LHBT = lactose hydrogen breath test; IBS = irritable bowel syndrome; NS = not significant.

**Table 3 nutrients-13-03833-t003:** Frequency of the HLA DQ2 and DQ8 haplotypes, of intra-epithelial lymphocytosis in the duodenal mucosa, and of allergy test for IgE-mediated cow’s milk allergy in the 314 study subjects, grouped according to the LHBT results as “Lactose tolerant”, “Lactose intolerant”, and “Lactose digesters/intolerant”, and in the IBS controls who did not report milk intolerance.

	Lactose Tolerant (*n* = 100) A	Lactose Intolerant (*n* = 178) B	Lactose Digesters/Intolerant (*n* = 36) C	IBS Controls (*n* = 102) D	*p*
Presence of DQ2 and/or DQ8 haplotype	32 of 52 (61%)	59 of 110 (53%)	5 of 14 (35%)	17 of 51 (33%)	All comparisons were NS
Presence of duodenal intra-epithelial lymphocytosis (IEL > 25/100 EC)	17 of 27 (64%)	35 of 61 (57%)	7 of 10 (70%)	NA	All comparisons were NS
Allergy test for IgE-mediated cow’s milk allergy	9 of 100 (9%)	5 of 178 (3%)	1 of 36 (3%)	4 of 102 (4%)	A vs. B < 0.05 A vs. C NS A vs. D NS B vs. C NS B vs. D NS C vs. D NS

Notes: Only a certain percentage of the whole study group with self-reported milk intolerance underwent HLA typing and duodenal histology studies. Histology data were not available in the IBS controls who did not report milk intolerance. LHBT = lactose hydrogen breath test; IBS = irritable bowel syndrome; IEL = intra-epithelial lymphocytes; EC = epithelial cells; NA = not available; NS = not significant.

**Table 4 nutrients-13-03833-t004:** Number of stool movements and severity of intestinal symptoms and overall symptoms in the 24 h following the cow’s milk and the rice milk DBPC challenges in the 42 patients with self-reported milk intolerance who tested negative at the H2-lactose breath test. Data are expressed as median and range.

	Cow’s Milk Challenge	Rice Milk Challenge	*p* Values
Number of bowel movements	3 (1–5)	1 (0–3)	*p* = 0.002 (Z = −3.04)
Abdominal pain severity (VAS score)	50 (10–70)	10 (0–50)	*p* = 0.0001 (Z = −3.58)
Bloating (VAS score)	60 (10–80)	35 (10–50)	*p* = 0.005 (Z = −2.98)
Urgency of evacuation	75 (25–80)	20 (0–30)	*p* = 0.0001 (Z = −3.92)
Overall symptoms (VAS score)	60 (20–80)	20 (0–50)	*p* = 0.0001 (Z = −3.82)

Notes: Stool frequency, abdominal pain, bloating, urgency of evacuation, and overall symptoms severity were evaluated by a 100 mm visual analog scale recorded by the patients, with 0 representing no symptoms.

## Data Availability

Not applicable.

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
