# Peer review of "Whole Cow’s Milk but Not Lactose Can Induce Symptoms in Patients with Self-Reported Milk Intolerance: Evidence of Cow’s Milk Sensitivity in Adults"

_nutrients, 2021, doi:10.3390/nu13113833_

Round 1

Reviewer 1 Report

Many subjects with self-reported milk intolerance are asymptomatic during a lactose breath testing. Otherwise many self reported lactose intolerant patients are not even lactose malabsorbers. This study confirms these clinical evidences and also suggests, by a prospective approach and a double blind challenge method, the potential role of whole cow's milk (apart from lactose) in eliciting digestive symptoms. 

This is an interesting and innovative study that deserves publication.

Author Response

To the Editor of Nutrients

Dear Editor,

Thank you for the opportunity to revise and to resubmit our paper entitled “Whole cow’s milk but not lactose can induce symptoms in patients with self-reported milk intolerance: evidence of cow’s milk sensitivity in adults”. We would underline that one of the co-Authors (Chiara Tinè, MD) had been excluded from the Authors list in the original submission, due to our secretary mistaken. She has been now correctly inserted in the revised version and her role was specified. 

We are glad in reading the positive Reviewers comments and we are proud to have the opportunity to publish our innovative study in Nutrients. We have evidences that cow’s milk protein intolerance has a relevant role in determining gastro-intestinal and extra-intestinal symptoms in adults. We hope that after the publication of our study, new researches will add data about this important topic.

Sincerely yours

Antonio Carroccio

REVIEWER 1

Many subjects with self-reported milk intolerance are asymptomatic during a lactose breath testing. Otherwise, many self-reported lactose intolerant patients are not even lactose malabsorbers. This study confirms these clinical evidences and also suggests, by a prospective approach and a double-blind challenge method, the potential role of whole cow's milk (apart from lactose) in eliciting digestive symptoms. 

This is an interesting and innovative study that deserves publication.

We thank the reviewer for her/his positive comments.

REVIEWER 2

The paper presents novel approach to self-reported milk intolerance, with significant clinical conclusion:  percentage of SRMI patients were not suffering from lactose intolerance.

I would recommend the paper for publication after some revisions:

  1. check all SRMI abbr. for instance in paragraph 4 SMRI is used
  2. Technical: the style of tables is not like in the template
  3. Technical: References: please like in template use 1. instead of 1)

We thank the reviewer for her/his positive comments and suggestions.

We checked all SRMI abbreviations and corrected them.

We used the template style through the entire text. Similarly, the references style was modified as suggested.

Reviewer 2 Report

The paper presents novel approach to self-reported milk intolerance, with significant clinical conclusion:   percentage of SRMI patients were not suffering from lactose intolerance.

I would recommend the paper for publication after some revisions:

  1. check all SRMI abbr. for instance in paragraph 4 SMRI is used
  2. Technical: the style of tables is not like in the tempelate
  3. Technical: References: please like in tempelate use 1. instead of 1)

Author Response

(The authors gave the same response as above.)
